# Metagenomic Approach Reveals the Second Subtype of PRRSV-1 in a Pathogen Spectrum during a Clinical Outbreak with High Mortality in Western Siberia, Russia

**DOI:** 10.3390/v15020565

**Published:** 2023-02-18

**Authors:** Nikita Krasnikov, Anton Yuzhakov, Taras Aliper, Alexey Gulyukin

**Affiliations:** Federal State Budget Scientific Institution “Federal Scientific Center VIEV”, 109428 Moscow, Russia

**Keywords:** porcine reproductive and respiratory syndrome, nanopore sequencing, metagenomics, neurological disorders

## Abstract

Porcine reproductive and respiratory syndrome virus (PRRSV) has a significant economic impact on pig farming worldwide by causing reproductive problems and affecting the respiratory systems of swine. In Eastern Europe, PRRSV-1 strains are characterized by high genetic variability, and pathogenicity differs among all known subtypes. This case study describes the detection of a wide pathogen spectrum, including the second subtype PRRSV-1, with a high mortality rate among nursery piglets (23.8%). This study was conducted at a farrow-to-finish farm in the Western Siberia region of Russia. Clinical symptoms included apathy, sneezing, and an elevation in body temperature, and during the autopsy, degenerative lesions in different tissues were observed. Moreover, 1.5 percent of the affected animals displayed clinical signs of the central nervous system and were characterized by polyserositis. Nasal swabs from diseased piglets and various tissue swabs from deceased animals were studied. For diagnostics, the nanopore sequencing method was applied. All the samples tested positive for PRRSV, and a more detailed analysis defined it as a second subtype of PRRSV-1. The results, along with the clinical picture, showed a complex disease etiology with the dominant role of PRRSV-1 and were informative about the high pathogenicity of the subtype in question under field conditions.

## 1. Introduction

Porcine reproductive and respiratory syndrome (PRRS) is an infectious swine disease associated with reproductive failure in sows, such as abortions and giving birth to stillborn and mummified fetuses. Other severe clinical problems of the syndrome are related to fever, anorexia, and acute respiratory disorders in all age groups. The disease inflicts major losses on productivity and swine health worldwide [1,2]. The etiological agent of the disease is the PRRS virus (PRRSV): a small, enveloped, positive-sense, single-stranded RNA virus. This virus belongs to the family Arteriviridae and is subdivided into two major species: Betaarterivirus suid 1 (the European type; PRRSV-1) and Betaarterivirus suid 2 (the North American type; PRRSV-2) [3]. The European type is more prevalent in the Russian Federation, and all three known subtypes (1, 2, and 3) of PRRSV-1 have also been detected in Russia [4,5].

PRRSV is also known to be one of the causative agents in the development of the porcine respiratory disease complex (PRDC). PRDC is a multifactorial and complex disease caused by a combination of infectious pathogens, environmental stressors, and differences in keeping and management practices [6]. *Mycoplasma* sp. likewise plays an important role and generally acts as an agent that causes secondary infection in PRRSV-affected pigs. The most common *Mycoplasma* species in such cases is *M. hyopneumonie* [7]. At the same time, *M. hyorhinis*, usually known as a commensal of the upper airways, is increasingly regarded as an infectious agent that can cause severe neurological disorders and polyserositis in swine [8,9].

In this case study, we describe the viral and bacterial complexes that were identified during an outbreak at a pig farm in Western Siberia. Due to the high mortality and unidentified disease etiology on this farm, we made the decision to apply the metagenomic approach in order to gain additional insights into this issue. The major objective of this work was to evaluate the pathogen spectrum during an infectious disease outbreak. The findings suggest that PRRSV-1 and *M. hyorhinis* play a dominant role in the pathogenesis of an infectious symptom complex.

## 2. Materials and Methods

### 2.1. Farm Description and Sample Collection

The study was conducted on a farm in the Western Siberia region of Russia in February 2022. The oldest farm building was constructed in the 1980s. The type of farm production was a single-site farrow-to-finish unit with a three-week batch production system. The herd size at the farm was about 85 thousand heads, including 5.5 thousand sows, 25 thousand weaned piglets, and 40 thousand finishing pigs. The genetic make-up of the herd was hybrid, namely an offspring of the F1 generation (Large White breed × Landrace breed) and inseminated by the Duroc breed.

The sows were kept in crates in the farrowing unit. Mechanical ventilation, as a combination of supply and exhaust fans with a set temperature of 18 to 25 °C for different age groups, was applied. The floor type was concrete. The housing of the piglet compartment included a heated floor plate, heat lamps, and additional bedding material for suckling piglets. The weaning age of the piglets was 28 days.

The feeding strategy included dry feed, mostly in the form of pellets. The extra feed has been provided since the fifth day of life. For suckling piglets, the extra feed was based on a pig milk replacer. Standard interventions and manipulations for all age categories included teeth clipping, tail docking, and iron injections.

The farm has been noted as diseased in the past, and clinical problems were mostly associated with respiratory and enteric disorders. The farm had been known to be PRRSV-positive since the end of the 1990s. During the past two years, sows were vaccinated using the inactivated vaccine VERRES-PRRS (“Vetbiochem”, Moscow, Russia). Among other infections, streptococcal, circoviral infections (PCV types 2 and 3), Glasser’s, and edema diseases were also reported in previous diagnostic farm reports.

During clinical examination, the following symptoms were observed: lethargy, body temperature elevation, and sneezing. In addition, approximately 1.5 percent of the affected animals displayed central nervous system symptoms such as locomotive disorders, uncoordinated movement, and head tilt. Clinical problems most affected the 40–60-day age group. The mortality rate in the nursery unit was 23.8%. The pre-weaning mortality rate and mortality in the fattening unit were 9.4% and 4.9%, respectively.

An autopsy of five freshly deceased piglets revealed the following pathological features: an enlarged spleen (splenomegaly), swollen inguinal lymph nodes, and kidney and lung lesions (Figure 1a–c). Some of them also had an accumulation of gas in the intestine lumen. At an additional autopsy of three forcedly killed piglets with neurological symptoms except for general tissue lesions, polyserositis was observed (Figure 1d).

Four types of pooled swabs were sampled for further analysis (Table 1).

### 2.2. Nucleic Acid Extraction, Library Preparation, and Sequencing

As a diagnostic method, third-generation nanopore sequencing was used following a protocol developed by PathoSense BV (Merelbeke, Belgium) [10]. The main protocol steps included nucleic acid extraction using the Quick-DNA/RNA viral kit (Zymo Research, Irvine, CA, USA), reverse transcription using the SuperScript IV Reverse Transcriptase (ThermoFisher Scientific, Waltham, MA, USA), enrichment by PCR (KAPA HiFi HotStart ReadyMix; Roche, Switzerland), the purification of amplicons with the magnetic beads (AMPure XP; Beckman Coulter, Brea, CA, USA). The quantity and quality were verified using a NanoDrop OneC spectrophotometer (ThermoFisher Scientific, Waltham, MA, USA). For the library preparation, the Rapid Barcoding Kit (RBK-004; ONT, Oxford, UK) was used, and further sequencing was performed on a MinION flow cell (R9.4.1; ONT, Oxford, UK).

Both ORF5 and ORF7 sequences were chosen for genotyping the detected PRRSV-1. Sanger sequencing and prior sample preparation were conducted as described previously [11], with several changes. For DNA extraction from the PCR mixture, the Cleanup Mini Kit was used in accordance with the manufacturer’s instructions. (Evrogen, Moscow, Russia).

### 2.3. Metagenomic and Phylogenetic Analyzes

Raw reads in fast5 file format were produced by MinKNOW. FASTQ files were generated and demultiplexed using a Guppy basecaller in the super-accurate basecalling setting (v. 21.10.4, ONT, Oxford, UK). Further bioinformatics analysis was performed on the Ubuntu 18.04 platform using bioconda channel programs. Quality scores were filtered with NanoFilt (v. 2.8.0; [12]). Reads with a q-score lower than 7 were omitted. Subsequently, host reads were removed after alignment to the Sus scrofa genome (GenBank accession number GCA_000003025.6) using graphmap (v. 0.5.2; [13]) and samtools (v. 1.6; [14]). For taxonomic assignments, Kraken2 (v. 2.1.2; [15]) was used. For visualization, KronaTools (v. 2.8.1; [16]) and Pavian [17] were used to generate taxonomic charts and flow diagrams. In addition, filtered sequences were analyzed using the BLASTn method (BLAST v. 2.13.0) with customized databases. The best hit (lowest e-value) was visualized using KronaTools (v. 2.8.1; [16]).

Phylogenetic dendrograms for the ORF5 and ORF7 sequences were plotted using the maximum likelihood method and the GTR model (MEGA 7.0) [18]. The robustness of the topology was evaluated by 1000 bootstrap replications.

## 3. Results

All four sample types were positive for PRRSV according to the nanopore sequencing method.

The swab from a pig with neurological disorders was used for the ORF5 and ORF7 sequencing of PRRSV by the Sanger method. The obtained sequences were deposited into the GenBank sequence database under accession numbers NV_2022_ORF5 OQ435279 and NV_2022_ORF7 OQ435280, respectively. The phylogenetic analysis revealed that the detected isolate (NV 2022) belonged to subtype 2 PRRSV-1 (Figure 2).

The sample composed of tissue swabs showed high species diversity, both in viral and bacterial reads. Among viral reads, there was a prevalence of Rotavirus C, subtype G6P [5] (26%), and other viral reads were related to the following genera: Astrovirus, Sapelovirus, Picobirnavirus, Bocaparvovirus, and Parvovirus. The bacterial composition mainly included *Campylobacter* sp. (13%), *Helicobacter* sp., *Escherichia* sp., *Spirochaeta* sp., and *Chlamydia suis*.

The nasal swab sample also indicated significant species diversity. Except for PRRSV reads, the sample was prevalent in Astrovirus. In a low abundance, it also contained genome fragments related to the following genera and species: Picobirnavirus, Bocaparvovirus, Parainfluenzavirus, Pestiviris K, and Influenza A virus. Among the bacteria species, the nasal microbiome was rich in *Mycoplasma hyorhinis* (38%), followed in abundance by *Pasteurella multocida* (7%), *Proteus* sp. (7%), *Glaesserella parasuis* (5%), *Campylobacter* sp. (4%), *Trueperella* sp. (1%), and others (Figure 3).

The sample composed of TBS was high in PRRSV. Among the viral reads, the sample also included several reads of Porcine Parainfluenza Virus 1, and bacterial reads were overrepresented by *Escherichia* sp. (over 60% of bacterial reads).

The sample from the animals with neurological disorders was rich in *M. hyorhinis* (over 75% of all bacterial reads). Other bacterial genera and species included *Pseudomonas*, *Bacillus*, and *Escherichia coli*, each constituting 1% or less of the total bacterial reads. Viral reads were mainly represented by the PRRSV (95% of all viral reads) and, in a small amount (1% of each), by *Astrovirus* sp. and Pestivirus C.

## 4. Discussion

PRRSV poses a big threat to the porcine industry worldwide and plays a pivotal role in the development of PRDC. In the presented case study, the virus was identified in all the samples. However, the read abundance was different in all the sample types. The reads were prevalent in TBS and swabs from piglets with neurological disorders; otherwise, nasal swabs from sick piglets and pooled tissue swabs from deceased animals contained fewer quantities of viral genome fragments. The second subtype of PRRSV-1 is currently widely distributed in Russia [19]. There is only one full sequenced genome of this subtype: WestSib13 (GenBank: KX668221.1). This strain was first isolated from an aborted fetus in 2013 from a farm in West Siberia, Russia, and was characterized by anorexia, dyspnoea, tremor, and a high mortality level that indicated it was an extremely invasive pathogenic agent [20]. The circulation of the second subtype has also been confirmed in Lithuania and Belarus [5]. In the experimental work by Stadejek et al., pigs infected by the second subtype of PRRSV-1 demonstrated different clinical signs. The symptoms varied from mild manifestations combined with a slightly elevated body temperature, including fever for a few days in the case of the Russian ILI6 strain, to high-grade fever, an increased respiratory rate, and conjunctival hyperaemia in the case of the BOR59 strain from Belarus [21]. At the same time, infection with the first subtype did not result in significant clinical signs, as evidenced by numerous studies [21,22,23]. However, the third subtype was considered to be a highly virulent strain [24,25].

In our case of the natural outbreak, the level of mortality among the most affected group of pigs was 23.8%, and we considered this parameter to be sufficiently high for the PRRSV-affected animals.

As the PRRSV destroys macrophages in various internal organs and harms a significant part of the pig’s immunity system, secondary infections by bacteria and other viruses are possible, which affect the respiratory system and cause devastating damage to the lung and other accompanying tissues. An atypical form of PRRS can be associated with neurological signs [26].

*M. hyorhinis* is usually considered a commensal of the upper airways and is often found in the respiratory tract of pigs. At the same time, some studies have shown that the bacteria may complicate the disease process and exacerbate the development of PRRS, causing a secondary infection in swine [9,27]. It is also known for causing polyserositis, pericarditis, and polyarthritis in weaned piglets [28,29]. Furthermore, in some recent works, *Mycoplasma* sp. has been considered a brain invader [30], and *M. hyorhinis* has even been suspected as a potential pathogen of the central nervous system (CNS) in swine [8].

The presented clinical picture with a demonstration of nervous signs in a part of the studied piglets in conjunction with a strong prevalence of *M. hyorhinis* in the nasal microbiome and, especially, in samples from animals with neurological disorders suggests that the correlation between this bacterium’s presence and the pathogenic effect are not accidental.

*M. hyorhinis* is closely connected with another detected pathogen, *Glaesserella (Haemophilus) parasuis*, which also plays a significant role in the enhancement of secondary infection in swine with PRRS [31]. Associations between PRRSV, *M. hyorhinis*, and *G. parasuis* have already been studied by Palzer A. et al., and they proved that these bacteria were more often detected in PRRSV-positive pigs [32,33]. Moreover, *G. parasuis* can cross the blood–brain barrier (BBB) by invading the brain microvascular endothelial cells and probably affecting them [34]. As such, there is no one possible reason behind the development of nervous signs in the presented case.

In addition, the sample from piglets with neurological disorders contained a few reads related to Pestivirus C: the causative agent of classical swine fever. Detailed bioinformatics analysis using BLASTn revealed close phylogenetic relations (99.5%) with the sequence of the vaccine isolate LK-VNIVViM (GenBank: KF739397.1). In Russia, this attenuated vaccine is a required preventive veterinary practice. Because of this fact, Pestivirus C can cross the interplacental barrier; therefore, the vaccine’s influence on the development and manifestation of nervous signs in the studied piglets cannot be excluded [35].

## 5. Conclusions

The detection of a large amount of PRRSV genome fragments in the swabs from the lower respiratory tract and in the sample from the animals with neurological symptoms, compared to the minimal number of other pathogen genomes, indicates the leading role of this virus in the presented clinical case. Moreover, the study has shown high pathogenic activity of the virus under field conditions.

Against the background of the immune cell destruction of diseased piglets by the PRRSV, the secondary infection of *M. hyorhinis* became the most important pathogenic factor in this case. Recently, the role of these bacteria in the development of polyserositis and arthritis in piglets, along with an increase in morbidity and mortality, has been noticed. In the presented case, the genomes of *M. hyorhinis* were detected in average quantities in the tissue swabs of piglets with neurological disorders (polyserositis was also found at autopsy) as well as in the high abundance of nasal swabs from sick piglets. This fact can shed light on a new pathogenic feature of *M. hyorhinis* that has been linked to a cause of neurological disorders.

In addition, nanopore sequencing, as a revolutionary molecular method, can be considered a novel solution for the search for new pathogenic complexes and may soon become a useful instrument for the diagnostics of infectious diseases.

## Figures and Tables

**Figure 1 viruses-15-00565-f001:**
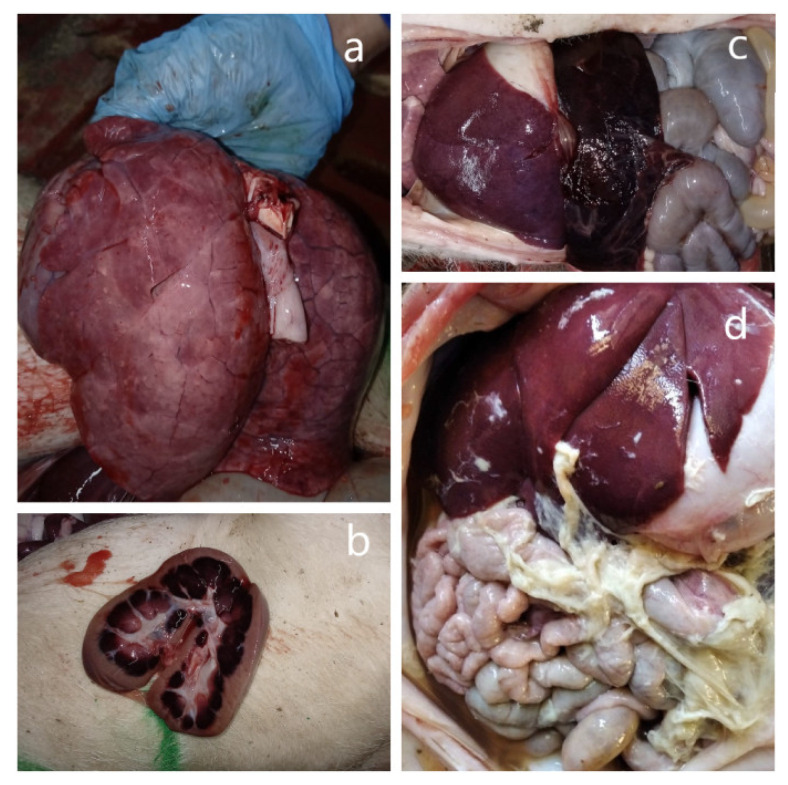
Macroscopic lesions in the affected organs during the autopsy of deceased animals. (**a**) Lungs. Multifocal petechial hemorrhages. (**b**) Kidney lesions. Large hemorrhages in medulla and petechiae in the cortex on the cut surface. (**c**) An enlarged, blood-filled spleen (splenomegaly). (**d**) Fibrinous polyserositis.

**Figure 2 viruses-15-00565-f002:**
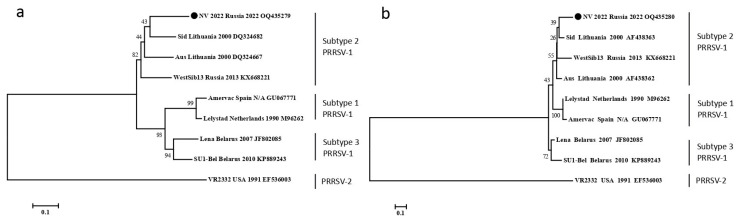
Phylogenetic trees of ORF5 nucleotide sequences (**a**) and ORF7 nucleotide sequences (**b**) of PRRSV strains. Multiple sequence alignments were obtained using the MUSCLE method. Bootstrap confidence limits are shown at each node. The NV 2022 isolate identified in this study is designated by a circle (●). The strain nomenclature is as follows: name, country of origin, year of isolation, GenBank accession number.

**Figure 3 viruses-15-00565-f003:**
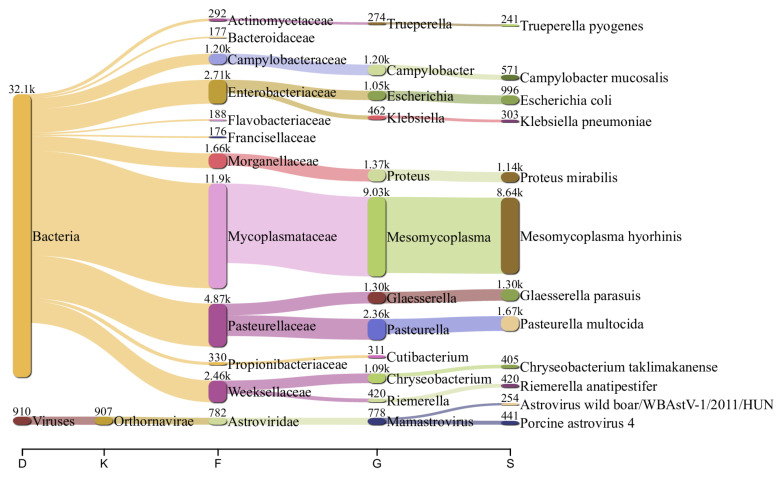
Flow diagram showing the most abundant viral and bacterial species (>100 reads) in the nasal swabs from sick piglets. The numbers above the nodes are the reads assigned to each taxon.

**Table 1 viruses-15-00565-t001:** Characteristics of the study samples for nanopore sequencing analysis.

Type of the Sample	Number and Type of the Animals
Tissue swabs (lung, spleen and kidney tissues, inguinal lymph node, and ileocecal valve swabs)	Five deceased piglets
Tracheo-bronchial swabs (TBS)	Three forcedly killed piglets
Nasal swabs	Five sick piglets
Tissue swabs from animals with neurological disorders (lung, spleen and kidney tissues, inguinal lymph node, and brain swabs)	Three forcedly killed piglets

## Data Availability

The datasets used and/or analyzed during the current study are available from the corresponding author on reasonable request.

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
