# Peer review of "Metagenomic Approach Reveals the Second Subtype of PRRSV-1 in a Pathogen Spectrum during a Clinical Outbreak with High Mortality in Western Siberia, Russia"

_viruses, 2023, doi:10.3390/v15020565_

Round 1

Reviewer 1 Report

I reviewed the manuscript entitled “Metagenomic approach reveals the second subtype of PRRSV-1 in a pathogen spectrum during a clinical outbreak with high mortality in Western Siberia, Russia”. In this manuscript authors present a case report involving high mortality in pigs, showing the relevance of metagenomic analysis in exemplifying the potential complex disease ethology in a high pathogenicity outbreak in pigs.

overall, I think it represents an interesting report. However, based on the way that the results are presented, the manuscript results very speculative. In the results authors show the presence of multiple bacterial and viral species present in the samples collected from the pigs in this farm, so that the title of this manuscript results very speculative. Instead, I would suggest something like: “Metagenomic approach reveals the complex disease ethology spectrum during a pig clinical outbreak with high mortality in Western Siberia, Russia”.

I consider that the introduction section should be improved to support the main goal of this research. In this context, I suggest using the information presented between lines 54 and 75, to give more context to this section. In this sense, the sentence between lines 48 and 51 should be improved to clarify the main goal of this research. I think, here should be stated that because of the high mortality and the unknown disease ethology in this farm authors decided to apply the metagenomic approach the get more insights into this problem.

On the other hand, the results section should be also improved. I suggest authors starting the results describing the clinical, and postmortem findings of this study. Use information between lines 76 and 94. Also, If authors wants to highlight the presence of some agents like PRRSV-1, claiming the existence of a second subtype, they should include a phylogenetic analysis to support this evidence.

Author Response

Reviewer  

I reviewed the manuscript entitled “Metagenomic approach reveals the second subtype of PRRSV-1 in a pathogen spectrum during a clinical outbreak with high mortality in Western Siberia, Russia”. In this manuscript authors present a case report involving high mortality in pigs, showing the relevance of metagenomic analysis in exemplifying the potential complex disease ethology in a high pathogenicity outbreak in pigs.

overall, I think it represents an interesting report. However, based on the way that the results are presented, the manuscript results very speculative. In the results authors show the presence of multiple bacterial and viral species present in the samples collected from the pigs in this farm, so that the title of this manuscript results very speculative.

Reviewer wrote: Instead, I would suggest something like: “Metagenomic approach reveals the complex disease ethology spectrum during a pig clinical outbreak with high mortality in Western Siberia, Russia”.

Answer: Thank you for the comment. But, we would like to highlight the PRRSv detection in the article title, as the virus was identified in all the samples both from diseased and deceased animals.

Reviewer wrote: I consider that the introduction section should be improved to support the main goal of this research. In this context, I suggest using the information presented between lines 54 and 75, to give more context to this section. In this sense, the sentence between lines 48 and 51 should be improved to clarify the main goal of this research. I think, here should be stated that because of the high mortality and the unknown disease ethology in this farm authors decided to apply the metagenomic approach the get more insights into this problem.

Answer: Thank you for the remark. We have expanded this part of the article, added information about the study goal, and substantiate the usage of the metagenomic approach in the considered case study.

This section now looks like this: In this case study, we describe the viral and bacterial complexes identified by the nanopore-based metagenomic approach during an outbreak of an infectious disease at a pig farm in Western Siberia. Due to the high mortality and unidentified disease etiology on this farm, we made the decision to apply the metagenomic approach in order to gain additional insight into this issue. The major objective of this work was to evaluate the pathogen spectrum during an infectious disease outbreak. The findings suggest that PRRSV-1 and M. hyorhinis play a dominant role in the pathogenesis of an infectious symptom complex.

Reviewer wrote: On the other hand, the results section should be also improved. I suggest authors starting the results describing the clinical and postmortem findings of this study. Use information between lines 76 and 94.

Answer: Thank you for your comment. Because the current manuscript is written as a case report, we suppose that the clinical and postmortem findings should be placed in the section with the farm description of the article.

Reviewer wrote: Also, If authors wants to highlight the presence of some agents like PRRSV-1, claiming the existence of a second subtype, they should include a phylogenetic analysis to support this evidence.

Answer: We agree with your opinion regarding including the phylogenetic analysis in the work. We conducted additional sequencing of the ORF5 and ORF7 fragments of the PRRSV genome using the Sanger method, deposited the sequences in GenBank, and constructed the phylogenetic trees. The phylogenetic analysis is presented in the Results section of the article.

Reviewer 2 Report

This study is very useful for reveal the property of second subtype of PRRSV-1. However, it will be better to rewrite Brief Report. Because, the reason of that the results of this study is not clear to reveal the property of PRRSV currently prevalent in western Siberia in Russia. But this study is very important to know the pathogenesis of PRRSV currently prevalent.

1.  Materials and Methods should be shortened follow the previous research.

2.  Figure 1 and Table 1 should be deleted.

Author Response

Reviewer wrote: This study is very useful for reveal the property of second subtype of PRRSV-1. However, it will be better to rewrite Brief Report. Because, the reason of that the results of this study is not clear to reveal the property of PRRSV currently prevalent in western Siberia in Russia. But this study is very important to know the pathogenesis of PRRSV currently prevalent.

  1. Materials and Methods should be shortened follow the previous research.
  2. Figure 1 and Table 1 should be deleted.

Answer: Thank you for the opinion regarding our manuscript. We consider the work a case report because, according to the journal's criteria, case reports present detailed information on the symptoms, signs, and diagnosis, as we described in the chapter with the farm description and methods in our manuscript. Case reports also highlight diagnostic approaches, and we consider the nanopore-based metagenomic approach as a modern diagnostic tool for the veterinary science. Overall, we leave this issue up to the editor's discretion. 

We suppose that all figures and tables are crucial to the manuscript. Figure 1 shows the pathological findings during necropsy, including organ lesions and polyserositis. Table 1 demonstrates all important characteristics of the studied samples, such as the sample composition, type and quantity of the studied animals. In addition, we have chosen the table format for data presentation because of its convenience. 

Reviewer 3 Report

The manuscript is a case report. It describes a Metagenomic Approach for the diagnostic of a clinical outbreak in a farrow-to-finish swine farm in the Western Siberia region of Russia. The analysis allowed the detection of the second subtype PRRSV-1 and a wide spectrum of pathogens.

Metagenomics is an innovative approach and in particular Nanopore sequencing permits to work on field. The subject of this paper is interesting not much for the findings, but for the approach applied. 

The work is well written and potentially significant, anyway, some important flaws should be resolved.

First of all, the authors should provide a more detailed description of material and methods for nucleic acid extraction, library preparation and sequencing (lines 96-105). These are fundamental steps of a metagenomic approach. The information provided not only are not sufficient for the reproduction of the method, but also what reported cannot be considered a mere metagenomic approach. The authors indeed give a reference for the details of the procedure (“Xie, J., Vereecke, N., Theuns, S., Oh, D., Vanderheijden, N., Trus, I., Sauer, J., Vyt, P., Bonckaert, C., Lalonde, C., Provost, C., 269 Gagnon, C.A., Nauwynck, H., 2021. Comparison of Primary Virus Isolation in Pulmonary Alveolar Macrophages and Four 270 Different Continuous Cell Lines for Type 1 and Type 2 Porcine Reproductive and Respiratory Syndrome Virus. Vaccines 9, 594. 271 https://doi.org/10.3390/vaccines9060594 “) that describes the Nanopore sequencing of PRRSv strains using a “PRRSV-specific enrichment prior to library preparation and sequencing”. This is not a metagenomic approach. The authors should share the true and detailed protocol they used or change the title of their work.

Moreover, the authors in the Results paragraph talk about bacterial reads and viral reads (lines 128-129, lines 136, 140). Did they use only one approach for sample and library preparation for bacteria, RNA viruses and DNA viruses?  

Besides, the authors report BLASTn results for their PRRSv sequence (line 152), Pestivirus C  (line 196), but they not provide the sequences used for these analysis. As the sequences obtained are important for the evaluation of their results and their work the authors should share them and register in a public database.

Author Response

Reviewer

The manuscript is a case report. It describes a Metagenomic Approach for the diagnostic of a clinical outbreak in a farrow-to-finish swine farm in the Western Siberia region of Russia. The analysis allowed the detection of the second subtype PRRSV-1 and a wide spectrum of pathogens.

Metagenomics is an innovative approach and in particular Nanopore sequencing permits to work on field. The subject of this paper is interesting not much for the findings, but for the approach applied.

The work is well written and potentially significant, anyway, some important flaws should be resolved.

Reviewer wrote:

First of all, the authors should provide a more detailed description of material and methods for nucleic acid extraction, library preparation and sequencing (lines 96-105). These are fundamental steps of a metagenomic approach. The information provided not only are not sufficient for the reproduction of the method, but also what reported cannot be considered a mere metagenomic approach. The authors indeed give a reference for the details of the procedure (“Xie, J., Vereecke, N., Theuns, S., Oh, D., Vanderheijden, N., Trus, I., Sauer, J., Vyt, P., Bonckaert, C., Lalonde, C., Provost, C., 269 Gagnon, C.A., Nauwynck, H., 2021. Comparison of Primary Virus Isolation in Pulmonary Alveolar Macrophages and Four 270 Different Continuous Cell Lines for Type 1 and Type 2 Porcine Reproductive and Respiratory Syndrome Virus. Vaccines 9, 594. 271 https://doi.org/10.3390/vaccines9060594 “) that describes the Nanopore sequencing of PRRSv strains using a “PRRSV-specific enrichment prior to library preparation and sequencing”. This is not a metagenomic approach. The authors should share the true and detailed protocol they used or change the title of their work.

Answer: Thank you for this important remark. The protocol we used was developed by the PathoSense BV Company and described briefly because it is private information, and we cannot disclose all the steps according to the personal request of the company. The reference was given merely for the mention of PathoSense BV in the manuscript. We did not use specific enrichment for any viral or bacterial species, including PRRSV, prior to the library preparation and sequencing. To avoid further misunderstandings, we have removed the reference from the manuscript text.

Reviewer wrote:

Moreover, the authors in the Results paragraph talk about bacterial reads and viral reads (lines 128-129, lines 136, 140). Did they use only one approach for sample and library preparation for bacteria, RNA viruses and DNA viruses?

Answer: Thank you for the clarifying question. Yes, we did use a single approach for sample, and library preparation for all nucleic acids in the samples. We did not use specific enrichment for any viral or bacterial species. In the article, we talk about reads in context of the bioinformatics analysis using special databases to perform classification of the detected species.

Reviewer wrote:

Besides, the authors report BLASTn results for their PRRSv sequence (line 152), Pestivirus C (line 196), but they not provide the sequences used for these analysis. As the sequences obtained are important for the evaluation of their results and their work the authors should share them and register in a public database.

Answer: We absolutely agree with you, and for PRRSv, we additionally performed sequencing of ORF5 and ORF7 sequences and deposited them in GenBank (the accession numbers are available in the Results section of the article). We suppose that the coverage of nanopore reads was not sufficient to generate a proper sequence. That is why we have decided to use additional Sanger sequencing in this case.

We are also ready to provide raw nanopore data upon any reasonable personal request. We have added this point to the Data Availability Statement section.

In the case of Pestivirus C, there were only 4 reads of this virus in the sample, and we could not even generate any possible contigs, but BLAST software helped us to understand that this virus was close to the vaccine strain.

Round 2

Reviewer 1 Report

I thank the authors for their responses. At this point I don't have more concerns about this manuscript. 

Reviewer 2 Report

This study was revised well according to my suggestions. 

Reviewer 3 Report

The authors took into account the reviewer’s suggestions.